# CNPY3 Promotes Human Breast Cancer Progression and Metastasis via Modulation of the Tumor Microenvironment

**DOI:** 10.3390/cimb47110883

**Published:** 2025-10-24

**Authors:** Xiaofeng Duan, Ran Zhao, Shaoli Sun, Beichu Guo, Zihai Li, Bei Liu

**Affiliations:** 1Department of Microbiology & Immunology, Hollings Cancer Center, Medical University of South Carolina, 86 Jonathan Lucas Street, Charleston, SC 29425, USA; xduan@tmu.edu.cn (X.D.); zhaoziyan91@126.com (R.Z.); bguo@allerdia.com (B.G.); 2Department of Pathology, The Ohio State University Wexner Medical Center, Columbus, OH 43210, USA; shaoli.sun@osumc.edu; 3Division of Medical Oncology, Department of Internal Medicine, The Ohio State University Comprehensive Cancer Center, Columbus, OH 43210, USA; zihai.li@osumc.edu; 4The Pelotonia Institute for Immuno-Oncology, The Ohio State University Comprehensive Cancer Center, Columbus, OH 43210, USA; 5Division of Hematology, Department of Internal Medicine, The Ohio State University Comprehensive Cancer Center, Columbus, OH 43210, USA

**Keywords:** CNPY3, molecular chaperone, therapeutic target, unfolded protein response, tumor microenvironment

## Abstract

Canopy FGF signaling regulator 3 (CNPY3) is a cochaperone of the molecular chaperone GRP94. CNPY3 is critical for the post-translational maturation of toll-like receptors and for regulating inflammasome signaling. However, the role of CNPY3 in cancer development and progression is still not fully understood. In this study, we aimed to investigate the role of CNPY3 in human breast cancer progression and metastasis. We used genomic and clinical information from multiple databases to profile CNPY3 and GRP94 in human cancers. We found that CNPY3 and GRP94 were elevated in human breast cancers compared to normal tissue. Higher expression of CNPY3 correlated with cancer progression and poor clinical outcomes in breast cancers. We confirmed these findings using a human breast cancer tissue array. We silenced CNPY3 in human breast cancer cells using a CRISPR/Cas9 system. For the first time, we found that deletion of CNPY3 significantly reduced tumor growth and metastasis in vitro and in vivo. Additionally, network and enrichment analyses revealed that changes in the unfolded protein response pathway and immune-related genes were significantly dependent on alterations in CNPY3 and GRP94. This study suggests that CNPY3 is a potential biomarker and novel therapeutic target for cancers.

## 1. Introduction

A broad range of stressful conditions caused by intrinsic and extrinsic factors may exist with the tumor development and progression, including hypoxia, changes in redox homeostasis, disrupted cellular metabolism, acidosis, and genetic lesions, leading to the production of mutated proteins, fast cell proliferation, and abnormal protein synthesis, which then induce endoplasmic reticulum (ER) stress and trigger the unfolded protein response (UPR) [1,2,3,4]. Then, UPR subsequently regulates downstream gene expression related to protein folding, such as molecular chaperones, which are specialized proteins that assist in the proper folding, assembly, and stability of other proteins (termed client proteins), preventing their misfolding or aggregation under cellular stress. Co-chaperones act as regulators that enhance chaperone specificity and efficiency by targeting specific client proteins or modulating chaperone activity. In the ER, chaperone–co-chaperone complexes are essential for maintaining proteostasis, and their dysfunction is linked to diseases, including cancer [2,5].

Previous studies have demonstrated that glucose-regulated protein 94 (GRP94, encoded by *HSP90B1* gene), also known as GP96, a stress-inducible molecular chaperone, is upregulated in various stress conditions that disrupt ER functions and homeostasis, including glucose starvation, oxidative stress, ER calcium store depletion, and the accumulation of misfolded proteins [6]. Hence, GRP94 may play a key role in regulating the balance between cancer cell viability and apoptosis through maintaining ER protein folding capacity and activating UPR sensors and ER-associated pro-apoptotic machinery. Moreover, previous studies have demonstrated that GRP94 is an essential molecular chaperone for toll-like receptors (TLRs), integrins, LRP6, GARP, and other vital innate receptors [7,8,9]. GRP94 and its client proteins are critical in oncogenesis and tumor development [10,11,12,13,14,15,16].

Previously, we discovered that canopy FGF signaling regulator 3 (CNPY3), also known as a protein associated with TLR4 (PRAT4A) [17], is a cochaperone of GRP94, which is critical for the post-translational maturation of TLRs [18]. CNPY3 is an ER resident protein initially involved in regulating TLR4 expression on the cell surface [19,20] and is required for multiple TLR trafficking and responses to TLR ligands except for the TLR3 ligand [17,21,22]. Interestingly, a recent study demonstrated that loss of CNPY3 in macrophages decreased activation of caspase-1 and processing of IL-1β and IL-18, which is independent of TLR signaling [23]. The deletion of CNPY3 altered the inflammasome structure and impaired ASC/caspase-1 colocalization [23], suggesting that CNPY3 might interact with ER molecules involved in ER stress responses and are critical in regulating the inflammasome. More recent studies showed that CNPY3 was upregulated in colon adenocarcinoma, and a higher level of CNPY3 was correlated with worse clinical outcomes. Knockdown of CNPY3 in colon tumor cell lines decreased tumor cell growth and migration in vitro [24]. However, the role of CNPY3 in cancer development and progression is still not completely understood.

Breast cancer is the second leading cause of malignant death in women. Despite the introduction of new early detection and treatments, breast cancer remains the development of multidrug resistance and metastasis. About 316,950 new cases of female breast cancer will be diagnosed in 2025 in the United States, with 14,600 deaths. Therefore, identifying targets for the development of novel therapeutics is still a major challenge. In this study, we demonstrated that expression levels of CNPY3 and GRP94 were elevated in human breast cancers, and a high level of CNPY3 was associated with an aggressive phenotype and poor clinical outcome. Furthermore, targeting CNPY3 in human breast cancer cells by CRISPR/Cas9 significantly reduced tumor growth and metastasis in vitro and in vivo. Mechanistically, we found that GRP94 and its co-chaperone CNPY3 regulate UPR pathway genes and immune-related genes in the tumor microenvironment. For the first time, our study uncovers the vital role of CNPY3 in the progression and metastasis of breast cancer, suggesting that CNPY3 may be a novel biomarker and therapeutic target for cancers.

## 2. Materials and Methods

### 2.1. Mice and Cell Lines

NSG mice were purchased from JAX Laboratory (Bar Harbor, ME, USA) and were bred and maintained according to the established guidelines and a protocol approved by the Medical University of South Carolina Institutional Animal Care and Use Committee. MCF-7 and MDA-MB-231 human breast cancer cell lines were kindly provided by Beichu Guo (Medical University of South Carolina). Human breast cancer cells were cultured in Eagle’s minimum essential medium (Sigma-Aldrich, St. Louis, MO, USA) supplemented with 10% heat-inactivated fetal calf serum (FCS, Atlas Biologicals, Fort Collins, CO, USA) and penicillin-streptomycin (Gibco, Waltham, MA, USA). All cells were cultured in a 5% CO_2_ incubator.

### 2.2. Reagents

Antibodies against CNPY3 were purchased from Sigma-Aldrich, and the GRP94 antibody was purchased from Enzo Life Sciences, Inc. (Farmingdale, NY, USA). The ABC-HPR kit and DAB substrate kit were obtained from Vector Laboratories (Newark, CA, USA). All other chemicals were obtained from Sigma-Aldrich and Fisher Scientific (Waltham, MA, USA).

### 2.3. Oncomine Database

The Oncomine Platform (Thermo Fisher, Ann Arbor, MI, USA; http://www.oncomine.org, accessed on 15 October 2025) provides translational bioinformatics solutions and multinational comparisons, along with clusters, gene expression signatures, and gene-set modules, as well as extracts biological insights from the data [25,26]. The Oncomine contains 715 datasets and 86,733 samples. Through an Oncomine Research Premium Edition Upgrade, we downloaded the raw datasets, including mRNA expression, clinical and pathological information, and survival data.

### 2.4. The cBioPortal for Cancer Genomics

cBioPortal (http://cbioportal.org, accessed on 15 October 2025) provides a resource with multidimensional cancer genomics datasets for cancer research, currently providing access to data from more than 48,511 tumor samples from 168 cancer studies [27].

### 2.5. Kaplan–Meier Plotter

Kaplan-Meier plotter (http://www.kmplot.com, accessed on 15 October 2025), an online database, including gene expression data and clinical data (breast cancer) [25], was further used to confirm the prognostic significance of the *CNPY3* and *HSP90B1* mRNA expression in breast cancers. Briefly, the *CNPY3* and *HSP90B1* genes were uploaded into the database to obtain Kaplan–Meier survival plots. Log-rank p-values and hazard ratios (HRs) with 95% confidence intervals (Cis) were calculated and displayed. The desired Affymetrix IDs of the *CNPY3* and *HSP90B1* genes were 217931_at and 200599_s_at, respectively. In this study, the patients were split by the best cutoff value of the gene expression level. We furt”er a’alyzed the prognostic value of *CNPY3* and *HSP90B1* gene expression in cancer subtypes and selected cohorts.

### 2.6. Tumor Tissue Array and Patient Data

Tissue microarrays (TMAs) for breast cancer specimens were obtained from US Biomax, Inc. (Rockville, MD, USA). Breast cancer cohort 1 (BC081120b) consisted of tumors and adjacent normal tissue from 110 patients, including tumor-node-metastasis (TNM) stage and pathology grade, estrogen receptor (ER), progesterone receptor (PR), and human epidermal growth factor receptor 2 (HER2) immunohistochemistry results. Breast cancer cohort 2 (BRM961) contained 48 cases of breast cancer with matched metastatic lymph nodes and adjacent normal tissue, including histological grade and TNM stage information.

### 2.7. Immunohistochemistry (IHC)

TMA series (5 μm thick) were deparaffinized and hydrated under the standard protocol. Then, we performed antigen retrieval using sodium citrate buffer and permeabilized slides with cold methanol at −20 °C for 5 min. For blocking endogenous peroxidase activity, the slides were incubated for 5 min at room temperature (RT) in 0.3% H_2_O_2_ PBS solution plus 0.3% normal goat serum. Tissues were blocked with 2% bovine serum albumin (BSA) and 10% normal goat serum in PBS at RT for 2 h. The slides were incubated in 1% BSA and 1% NGS/PBS for 1 h at RT with a rabbit anti-human CNPY3 antibody (Atlas Antibodies, Inc., Stockholm, Sweden) and a rat anti-GRP94 antibody (9G10, Enzo Life Sciences, Inc.), followed by blotting with biotin-anti-rabbit or biotin-anti-rat IgG antibody for 30 min (ABC kit, Vector Lab, Inc.) at RT. After washing, the staining signal was developed by 3,3′-diaminobenzidine tetrahydrochloride (DAB Kit, Vector Lab, Inc.) and counterstained with hematoxylin. The slides were scored blindly by Dr. Shaoli Sun, an experienced pathologist. The criteria are: 0 (negative staining), 1 (weak staining), 2 (moderate staining), 3 (strong staining), or 4 (very intense staining). The TMA slides were scanned with an Olympus BX61 microscope.

### 2.8. CNPY3 CRIPR/Cas9 Viral Vectors and Transduction

The human *CNPY3* CRISPR/Cas9 lentiviral vector and control vector were purchased from Open Biosystems (Huntsville, AL, USA). *CNPY3*-CRISPR/Cas9 lentivirus was packaged in the 293FT cells. For deletion of *CNPY3*, human breast cancer cells (1 × 10^5^/well) were seeded in a 12-well plate, then spin-infected with recombinant lentivirus, and centrifuged at 3000 rpm at 32 °C for 90 min.

### 2.9. Protein Extraction and Western Blot

Protein extraction and immunoblotting were performed as described previously [10]. Briefly, cells were washed three times with ice-cold PBS and lysed in radioimmunoprecipitation assay (RIPA) lysis buffer (0.01 M sodium phosphate, pH 7.2, 150 mM NaCl, 2 mM EDTA, 1% NP-40, 1% sodium deoxycholate, 0.1% SDS, and protease inhibitor cocktail). Total cell lysates were separated by 10% to 12% SDS-PAGE and transferred onto Immobilon-P membranes. Then, the membrane was blocked with 5% nonfat milk in PBS and blotted with different antibodies. Protein bands were visualized by using enhanced chemiluminescent (ECL) substrate (Thermo Scientific, Waltham, MA, USA).

### 2.10. Colony Formation Assay

Colony formation assays were performed as described previously. Briefly, Seed 2 × 10^3^ WT control and pool CNPY3 knockout (KO) MCF-7 and MDA-MB-231 human breast cancer cells per well in a 6-well plate and changed the medium the next day. Until the clones were visible, the cells were rinsed twice with PBS, fixed with methanol for 5 min, and washed twice with PBS. Then, the cells were stained with 0.5% crystal violet for 10 min and washed twice with PBS. The colonies were counted.

### 2.11. Human Breast Cancer Xenograft Model

Eight- to twelve-week-old NSG mice were implanted with 5 × 10^6^ WT control and CNPY3 KO MDA-MB-231 human breast cancer cells into the fourth mammary gland. The tumor growth was measured every three days using digital calipers [measuring the length (mm) and width (mm)]. Tumor volume {V = [(width)^2^ × length]/2, mm^3^} was plotted. We also monitored endpoint survival and general health indicators, including overall behavior, feeding, neuromuscular tone, body weight, and appearance of fur. At the endpoint, the primary tumor was excised and weighed, and metastatic tumors in the liver and lymph nodes were examined under a dissecting microscope after the mice were sacrificed.

### 2.12. Statistical Analysis

All statistical analyses were performed using the SPSS 24.0 (SPSS Inc., Chicago, IL, USA) and GraphPad Prism 9 (GraphPad Software, Inc., San Diego, CA, USA) statistical software packages. An unpaired two-tailed Student’s *t*-test or Mann–Whitney test was used to compare the differences between groups. Survival curves were plotted using the Kaplan–Meier method. The receiver operating characteristic (ROC) curve was used to determine the optimal cutoff point of gene expression in low-risk and high-risk patients. A *p*-value of less than 0.05 was considered to be statistically significant (* *p* < 0.05; *** p* < 0.01; **** p* < 0.001; **** *p* < 0.0001).

## 3. Results

### 3.1. The Expression of CNPY3 and GRP94 Is Upregulated in Human Cancers

Previous studies have shown that GRP94 is highly expressed in tumor cells [9,10,13]. However, it is unclear whether its co-chaperone, CNPY3, plays a critical role in tumor progression. We first investigated the CNPY3 and GRP94 expressions in 20 human cancers by the Human Protein Atlas. CNPY3 and GRP94 were overexpressed in most human cancers compared with normal tissue (Appendix A). Additionally, we found that the most common cancer types with increased CNPY3 and GRP94 were breast, colon, and gastric cancers (Appendix A). These results indicate that CNPY3 may play a crucial role in cancer development in the stressful TME.

### 3.2. CNPY3 and GRP94 Are Hallmarks of Human Breast Cancer Development and Progression

To validate the coordination of *CNPY3* and *HSP90B1* to predict an aggressive phenotype, we next determined the correlation of these two genes with clinical parameters based on the Curtis breast dataset (n = 2136) downloaded from the Oncomine database [28] (Appendix A). Both *CNPY3* and *HSP90B1* were overexpressed in breast cancers (Figure 1A). Also, we confirmed the consistency in the TCGA breast cancer dataset (n = 593) and Ki colon dataset (n = 123) (Appendix A, Appendix A). Furthermore, high expression of *CNPY3* and *HSP90B1* correlated with poor differentiation (Figure 1B), PR negative status (Figure 1C), triple-negative breast cancer (TNBC) (Figure 1D), a high Nottingham Prognostic Index (NPI) (Figure 1E), and an aggressive basal subtype (Figure 1F). Additionally, we used the Kaplan–Meier plotter database to confirm the consistent predictive value for clinical outcomes. In a large group of breast cancer patients (n = 1907), patients with high levels of *CNPY3* and *HSP90B1* expression had shorten overall survival (Figure 2A) and progression-free survival (Figure 2B).

To determine whether protein expression levels of CNPY3 and GRP94 are consistent with gene expression levels and associated with clinical features in human breast cancers, we used the tissue microarray (TMA) approach, which included breast cancer, their corresponding adjacent normal tissue (ANT), and metastasis tissue (n = 158). For the first time, we found that CNPY3 was significantly overexpressed in cancer and metastatic tissues but not in ANT (Figure 3A, top and Figure 3B, left). Consistent with CNPY3, GRP94 was highly expressed in cancer and metastasis tissues compared with ANT (Figure 3A, bottom and Figure 3B, right). However, there is no significant difference in CNPY3 and GRP94 expression between primary and metastatic cancers. These results suggested that CNPY3 may be critical in human cancer development and progression.

### 3.3. Deletion of CNPY3 Suppresses Human Breast Tumor Growth and METASTASIS In Vitro and In Vivo

To further investigate the role of CNPY3 in breast cancer progression and metastasis, we next silenced CNPY3 in human breast cancer cell lines MCF-7 (ER^+^PR^+^BC) and MDA-MB-231 (TNBC) using a CRISPR/Cas9 lentiviral system. We confirmed that CNPY3 was completely knocked out in both MCF-7 and MDA-MB-231 tumor cells from different passages (Figure 4A). First, we performed tumor cell colony formation in vitro. We found that silencing of CNPY3 causes significantly compromised MCF-7 and MDA-MB-231 tumor cell growth compared with control tumor cells (Figure 4B). The MDA-MB-231 cell line is a poorly differentiated triple-negative breast cancer and has a stronger invasive ability than MCF-7, and it is a well-known metastatic breast cancer model. To determine whether CNPY3 is required for human breast cancer cell growth and metastasis in vivo, we performed a human breast cancer xenograft model by implanting WT control and CNPY3 KO MDA-MB-231 breast cancer cells into the fourth mammary gland in NSG mice and monitored tumor growth, endpoint survival, and metastasis. We found that deletion of CNPY3 in human breast cancer cells significantly decreased tumor growth and prolonged survival (Figure 4C,D) and reduced metastasis (Figure 4E), suggesting that CNPY3 may be a potential therapeutic target for cancers.

### 3.4. CNPY3 and GRP94 Alterations Regulate UPR Pathway Genes and Immune-Related Genes in the Tumor Microenvironment

GRP94 is a downstream molecule in the ER to mediate UPR [29]. It also plays a key immunological role through variety of client proteins [6]. A recent study showed that GARP, one of GRP94 client protein, promotes oncogenesis in breast cancer [11]. To understand the functional significance of CNPY3, a co-chaperone of GRP94, in the tumor microenvironment, we examined the co-expression of *HSP90B1* with co-chaperones, UPR pathway genes, lymphocyte markers, and cytokines using the Oncomine database and found that higher expression of *HSP90B1*, co-chaperone *CNPY3*, UPR pathway, and immune-related genes in common solid tumors (Figure 5A). In addition, we generated the network of *HSP90B1* and found that it is associated with 9 most frequently altered neighbor genes, including *CNPY3, ATF4, STAT3, TLR7, TLR8* and *TLR9* based on the cBioportal database (Figure 5B). However, we found a weak correlation between *HSP90B1* and *CNPY3* (Figure 5C) and *TLR7* (Figure 5D). There is no correlation of *HSP90B1* with *TLR8* (Figure 5E) based on the breast cancer microarray datasets [28]. The gene levels of CNPY3 and GRP94 might not be the same as protein levels. Whether CNPY3’s oncogenic effects are dependent on GRP94 and whether CNPY3 is correlated with GRP94 still need further investigation. To determine whether *HSP90B1* and *CNPY3* gene mutations impact the UPR, lymphocyte markers, and cytokines, we assessed gene mutations of *HSP90B1* and *CNPY3* with mutation rates of 4% and 5%, most of which were amplified and upregulated (Figure 5F). Next, based on the presence or absence of the *HSP90B1* mutation, we evaluated the mRNA expression of targeted genes and found that the *HSP90B1* gene mutation was associated with alterations in the expression of various genes, including genes in the UPR pathway (*ATF4*, *XBP1*, *CHOP, PDI*); Th17 and Th2 T cell markers (*RORC* and *GATA3*); and cytokines (*CCR6*, *CCL20*, *CXCL12*, *IL2*, *TNF*). Additionally, we found that these genes had a relatively high mutation rate of 4-26% (Figure 6A). Furthermore, we pursued the pairwise association analysis for *HSP90B1*, *CNPY3* and UPR and immune-related genes and confirmed the co-occurrence of the genomic alterations (Figure 6B). These genes were found to be significant prognostic factors based on Kmplot data (Figure 6C). Our study suggested that CNPY3, a co-chaperone of GRP94, may play important roles in regulating the tumor microenvironment and anti-tumor immune response.

## 4. Discussion

As an ER chaperone involved in protein quality control, stress, and inflammation, GRP94 promotes tumor progression through its role in folding a variety of clients [11]. Using a genetic strategy, our group has discovered that GRP94 is an essential chaperone for folding the Wnt co-receptor LRP6 [30] and that it is required for multiple myeloma cell survival, which is mediated in part by the Wnt target survivin [10]. Our group also demonstrated that GRP94 chaperones GARP [9], and the GARP-TGFβ axis plays an important role in platelets, which inhibit anti-tumor immunity. Platelet-specific deletion of the GARP blunted TGFβ activity in the TME and enhanced protective immunity against both melanoma and colon cancer [31]. Recent studies have shown that human epidermal growth factor receptor-2 (HER2) expression is elevated in approximately 18–25% of breast cancers [32], and GRP94 is required for both the proper intracellular trafficking and stability of HER2 protein [33]. Also, GRP94-specific inhibitors affected high-density HER2 formations at the plasma membrane, which is critical for proper HER2 signaling in breast cancers [34]. These GRP94 clients have been described to function at various stages of cancer development. Beyond its known function as a TLR-specific co-chaperone for GRP94 in the ER [17,18], CNPY3 regulates inflammasome activation by promoting ASC/caspase-1 colocalization independently of TLR signaling [23]. Furthermore, CNPY3 is upregulated in colon adenocarcinoma, where its expression correlates with poor clinical outcomes and promotes tumor cell proliferation and migration in vitro [24].

However, the role of CNPY3 in human cancers is still not fully understood. In this study, we pursued comprehensive molecular profiling of CNPY3 and GRP94 and found that both CNPY3 and GRP94 are upregulated in different human cancers, including breast, liver, colon, pancreatic, ovarian cancers, etc., suggesting that CNPY3 may be a potential biomarker and therapeutic target for pan cancers (Appendix A). Human breast cancer is a heterogeneous cancer with different types based on origins, molecular features, and responses to therapeutics. We found that the elevated expression of both CNPY3 and GRP94 was associated with an aggressive phenotype and poor clinical outcome (Figure 1, Figure 2 and Figure 3). Both CNPY3 and GRP94 overexpression were correlated with TNBC, advanced stage, high NPI, and poorly differentiated breast cancers (Figure 1). Additionally, both CNPY3 and GRP94 had consistent prognostic value for overall survival and progression-free survival in breast cancer (Figure 2). Intriguingly, targeting CNPY3 in human ER^+^PR^+^ and TNBC cells by CRISPR/Cas9 significantly decreased tumor growth, prolonged survival, and reduced metastasis in vitro and in vivo (Figure 4). We have shown that the expression level of CNPY3 is strongly associated with TNBC and basal subtypes (Figure 1), suggesting CNPY3 subtype-specific roles. Further studies comparing CNPY3 dependency across a broader panel of breast cancer subtypes will be a focus of our ongoing research. Clinically, TNBC is an aggressive breast cancer with a high rate of recurrence and is ineffective in responding to hormone and HER2-targeted therapies, as well as checkpoint blockade immunotherapy. This study may provide a foundation for the development of CNPY3-targeted therapy to overcome drug resistance and enhance immunotherapy. However, there are no small molecule inhibitors and antibodies available for targeting CNPY3. Further development of CNPY3-specific inhibitors for preclinical study is needed.

To identify potential oncogenic mechanisms underlying the elevated expression of CNPY3 and GRP94, we performed network analysis and found the nine most frequently altered neighboring genes, including *STAT3*, *ATF6*, *CNPY3*, *TLR7*, *TLR8*, and *TLR9* (Figure 5). As a protective factor, ATF6 orchestrates cellular adaptation to ER stress by fine-tuning the expression of UPR target genes, such as GRP78 and GRP94. A wide range of stressful conditions exist in tumor development and progression, and chaperones are needed to regulate the balance between cancer cell viability and apoptosis. Prolonged activation of ER stress sensors enables malignant cells to acquire enhanced tumorigenic, metastatic, and therapy-resistant capabilities [35]. STAT3 is a member of the STAT protein family. Triggered by diverse cytokines and growth factors, STAT3 modulates the expression of stimulus-responsive genes, thereby exerting a pivotal regulatory influence on cellular processes, including cell growth and apoptosis, invasion, and immunosuppression [36].

In this study, we also demonstrate that CNPY3 and GRP94 are associated with the UPR pathway and immunity-related genes, including lymphocyte cell markers and cytokines (Figure 5). After enrichment and pairwise co-occurrence analysis, we found that GRP94 and CNPY3 mediated changes in the expression of genes, including the UPR pathway genes *ATF4*, *XBP1*, *CHOP*, and *PDI*, and the immune-related genes *RORC*, *GATA3*, *IL2*, *TNF*, *CCR6*, *CXCL12*, and *CCL20* (Figure 6). Upon antigen stimulation through their T cell receptor, naive CD4^+^ T cells differentiate into distinct effector lineages, including Th1, Th2, and IL-17-producing Th17 cells. The commitment to distinct T helper (Th) lineages is orchestrated by lineage-defining transcription factors: T-bet governs Th1 differentiation, GATA3 dictates Th2 differentiation, and RORC drives Th17 cell fate specification [37,38]. Human Th17 cells express high levels of CCR6 and CXCR4. These homing receptors likely orchestrate Th17 cell migration and retention in inflammatory foci and tumor sites [39]. For instance, human tumor microenvironments exhibit elevated expression of CXCL12 (CXCR4 ligand) and CCL20 (CCR6 ligand), creating a chemokine gradient that potentiates Th17 cell infiltration into tumors. [39]. Furthermore, Th17 cells are established as pivotal regulators across multiple immune-mediated inflammatory diseases and cancer contexts [40]. Patients with high expression of the Th17 cluster had a poor prognosis [41]. CXCL12-CXCR4 signaling is implicated in immune cell tumor trafficking and tumor cell biology. CXCL12-CXCR4 signaling mediates plasmacytoid DC trafficking into tumors and Treg cell homing to the bone marrow microenvironment and is involved in tumor cell proliferation, metastasis, and tumor vascularization [39]. Hence, on the one hand, CNPY3 and GRP94 may promote prosurvival signaling through the UPR pathway to promote tumor cell survival under endogenous and exogenous stress conditions; on the other hand, they also play important immunosuppressive roles through cytokines to misbalance immune cells in the tumor microenvironment. However, how the CNPY3-UPR signaling pathway regulates the tumor microenvironment is still not clear. Further study is needed to understand how CNPY3 regulates the tumor microenvironment and anti-tumor immunity for the development of combination therapies for cancers.

## 5. Conclusions

Our study demonstrated that CNPY3, a cochaperone of immune chaperone GRP94, was elevated in the majority of human cancers, and the overexpression of CNPY3 and GRP94 predicts an aggressive phenotype and poor clinical outcome. Targeting CNPY3 in human breast cancer cells reduced tumor growth and metastasis. In addition, the changes in the UPR pathway and immune-related genes in the tumor microenvironment were significantly dependent on alterations in CNPY3 and GRP94. For the first time, our study uncovers the critical roles of CNPY3 in the progression and metastasis of breast cancer, suggesting that CNPY3 may be a novel biomarker and therapeutic target for cancers. However, the underlying mechanism of CNPY3 in regulating the tumor microenvironment and anti-tumor immunity needs to be further investigated.

## Figures and Tables

**Figure 1 cimb-47-00883-f001:**
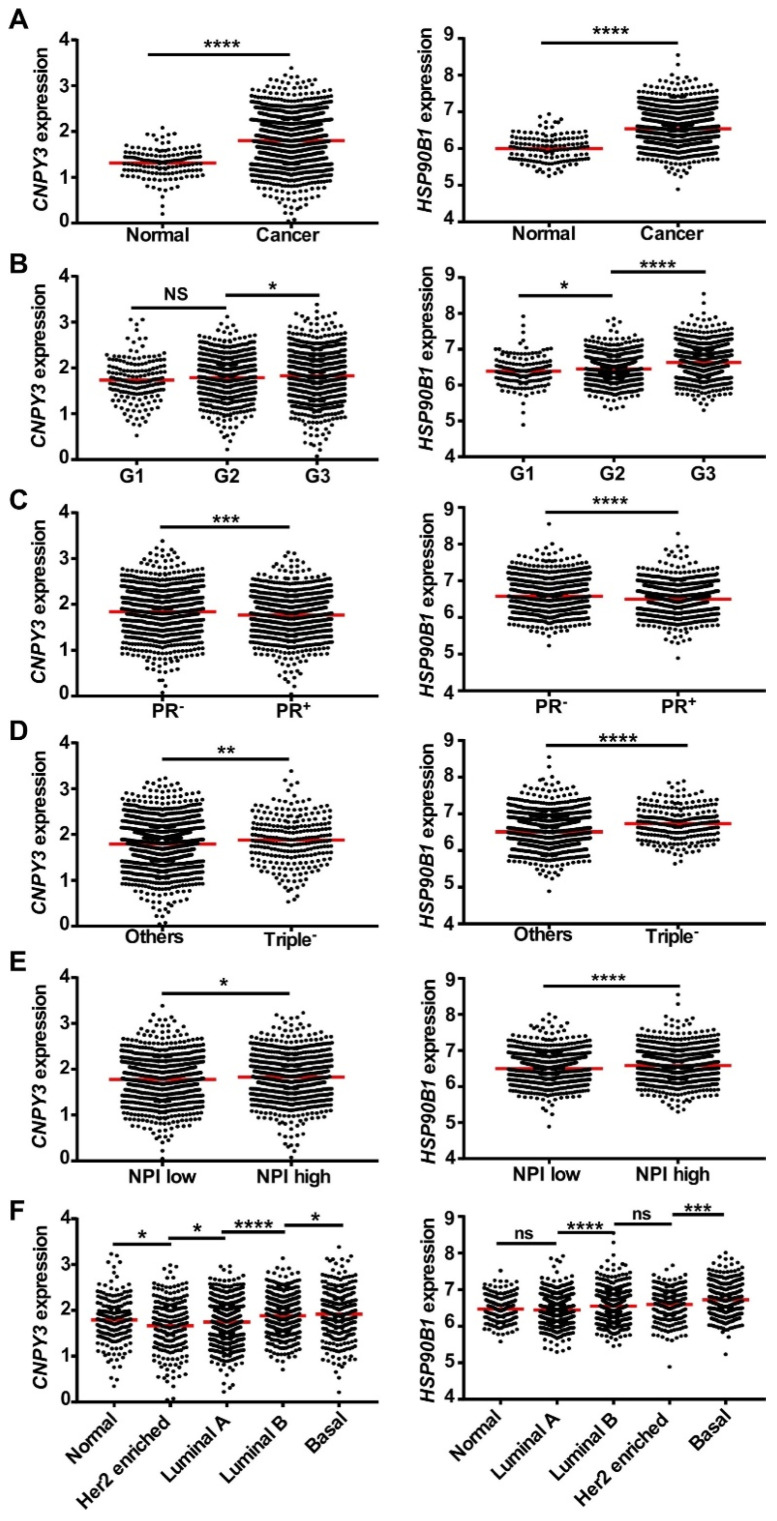
**High expression of*****CNPY3 and HSP90B1*****are correlated with disease progression phenotype in human breast cancers.** *CNPY3 and HSP90B1* predicts aggressive behavior in breast cancer based on 1980 breast cancer samples and 144 normal breast samples in Curtis et al. Nature dataset. (**A**) *CNPY3 and HSP90B1* mRNA were both overexpressed in breast cancers compared with adjacent normal tissues. (**B**–**F**) High mRNA levels of *CNPY3 and HSP90B1* correlate with poor pathological differentiation breast cancers (**B**), progesterone receptor (PR)-negative breast cancers (**C**), triple-negative breast cancers (**D**), high Nottingham Prognostic Index (NPI) breast cancers (**E**), and PAM50 luminal A, luminal B and basal subtype breast cancers (**F**). * *p <* 0.05, ** *p* < 0.01, *** *p* < 0.001, **** *p* < 0.0001, NS/ns: not significant.The red dot line represnts mean.

**Figure 2 cimb-47-00883-f002:**
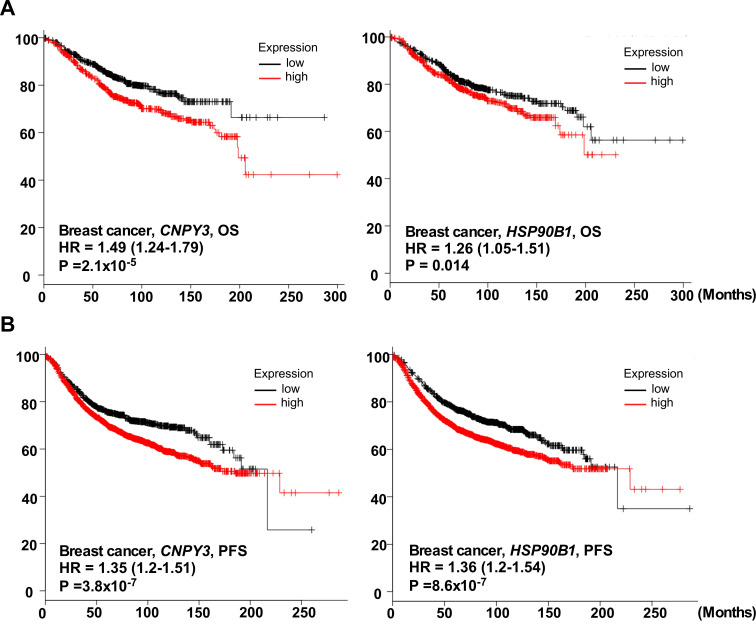
**Upregulation of*****CNPY3 and HSP90B1*****is associated with poor prognosis in human breast cancers.** (**A**) Breast cancer patients with high expression of *CNPY3 (***left***)* and *HSP90B1* (**right**) have significant poor overall survival (OS). (**B**) Breast cancer patients with high expression of *CNPY3 (***left***)* and *HSP90B1* (**right**) have significant poor progression-free survival (PFS). HR: hazard ratio.

**Figure 3 cimb-47-00883-f003:**
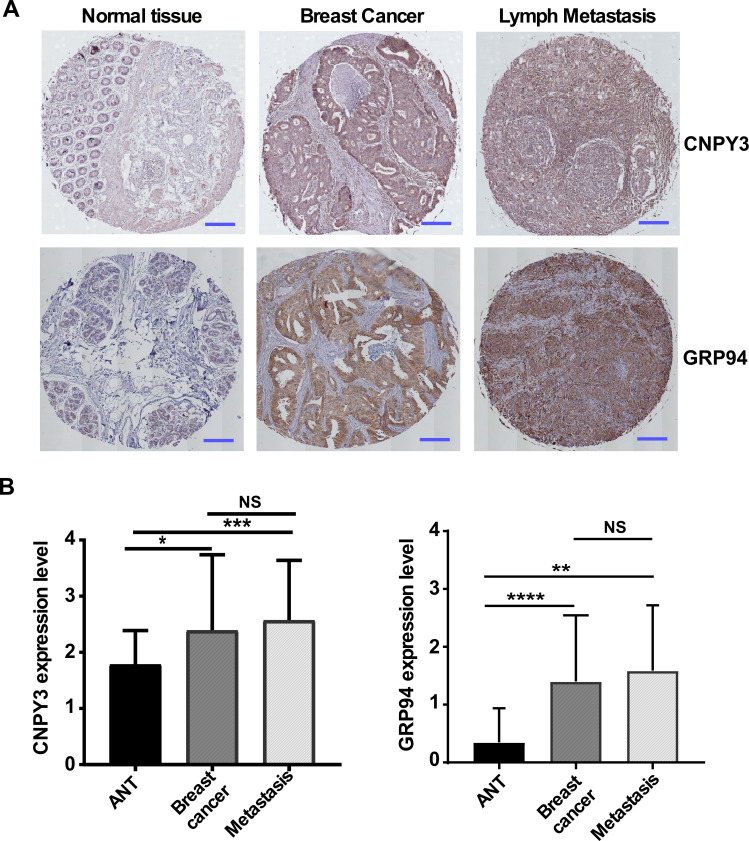
**CNPY3 and GRP94 are hallmarks of human breast cancer development and progression.** (**A**) Representative CNPY3 (top) and GRP94 (bottom) staining in adjacent normal tissue (ANT), breast cancer and lymph node metastasis tissue by immunohistochemistry. Scale bar: 25 μm. (**B**) Quantification of CNPY3 (**left**) and GRP94 (**right**) expression in breast cancer. n = 158 for ANT, tumor, and metastasis. NS: not significant. Data are shown as mean ± SD. * *p <* 0.05, ** *p* < 0.01, *** *p* < 0.001, **** *p* < 0.0001, NS: Not Significant.

**Figure 4 cimb-47-00883-f004:**
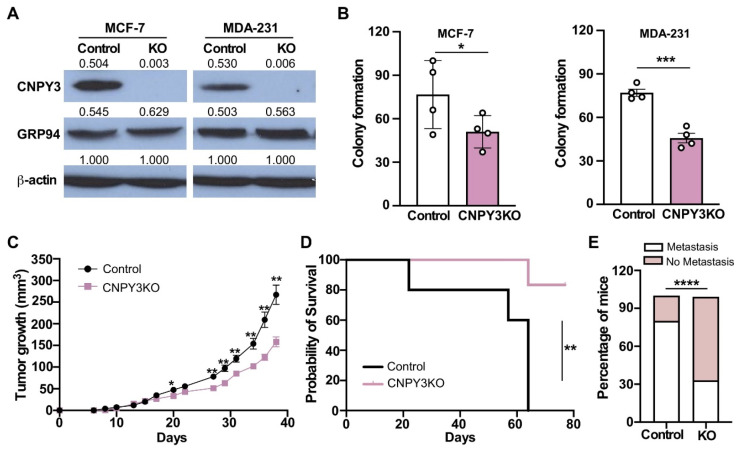
**Deletion of CNPY3 decreased human breast cancer cell growth and metastasis in vitro and in vivo.** (**A**) Immunoblot analysis of CNPY3, GRP94, and β-actin in empty vector (EV) control and CNPY3 knockout (KO) MCF-7 and MDA-231 human breast cancer cells. (**B**) Colony formation assays were performed to assess the control and CNPY3 KO MCF-7 and MDA-231 tumor cell proliferation. (**C**) Tumor growth of WT control and CNPY3 knockdown MDA-231 cells implanted into the fourth mammary gland in NSG mice (n = 6). (**D**) Kaplan–Meier curves for the endpoint survival after implantation of control and CNPY3 KO MDA-231 cells in NSG mice (n = 6). (**E**) Percentage of mice with metastasis and without metastasis. Error bars indicate SD. * *p <* 0.05, ** *p* < 0.01, *** *p* < 0.001, **** *p* < 0.0001.

**Figure 5 cimb-47-00883-f005:**
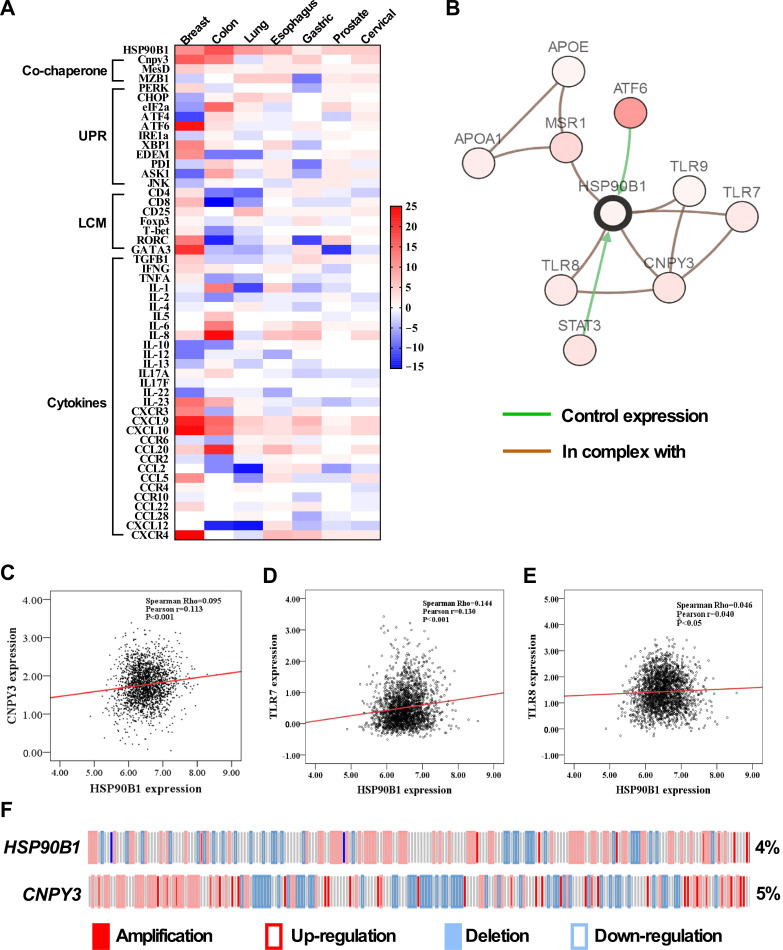
**Heatmap and network of*****HSP90B1*****and*****CNPY3*****in human cancers**. (**A**) Heatmap of mRNA expression for *HSP90B1*, co-chaperone, unfolded protein response (UPR) pathway, lymphocyte marker (LCM), and immune-related genes in common solid tumors. The red square indicates upregulation, and blue indicates downregulation, and *p* value < 0.05 (Oncomine Platform, Thermo Fisher, Ann Arbor, MI, USA). (**B**) *HSP90B1* and *CNPY3* gene network in cBioPortal for Cancer Genomics (http://www.cbioportal.org, accessed on 15 October 2025). The network contains 10 nodes, including *HSP90B1* and the 9 most frequently altered neighbor genes. (**C**) The weak correlation of *HSP90B1* and *CNPY3* mRNA levels in the Curtis et al., Nature 2012 dataset [28] (n = 2136, r = 0.113, *p* value < 0.001). (**D**) The weak correlation of *HSP90B1* and *TLR7* mRNA levels in the Curtis et al., Nature 2012 dataset (n = 2136, r= 0.130, *p* value < 0.001). (**E**) There was no correlation of *HSP90B1* and *TLR8* mRNA levels in the Curtis et al., Nature 2012 dataset (n = 2136, r = 0.040, *p* value < 0.001). (**F**) The gene alternation of *HSP90B1* and *CNPY3* in the Curtis et al., Nature 2012 dataset. Red solid bars indicate gene amplifications, blue solid bars are deletions, and red blank bars indicate upregulation, and blue blank bars indicate downregulation.

**Figure 6 cimb-47-00883-f006:**
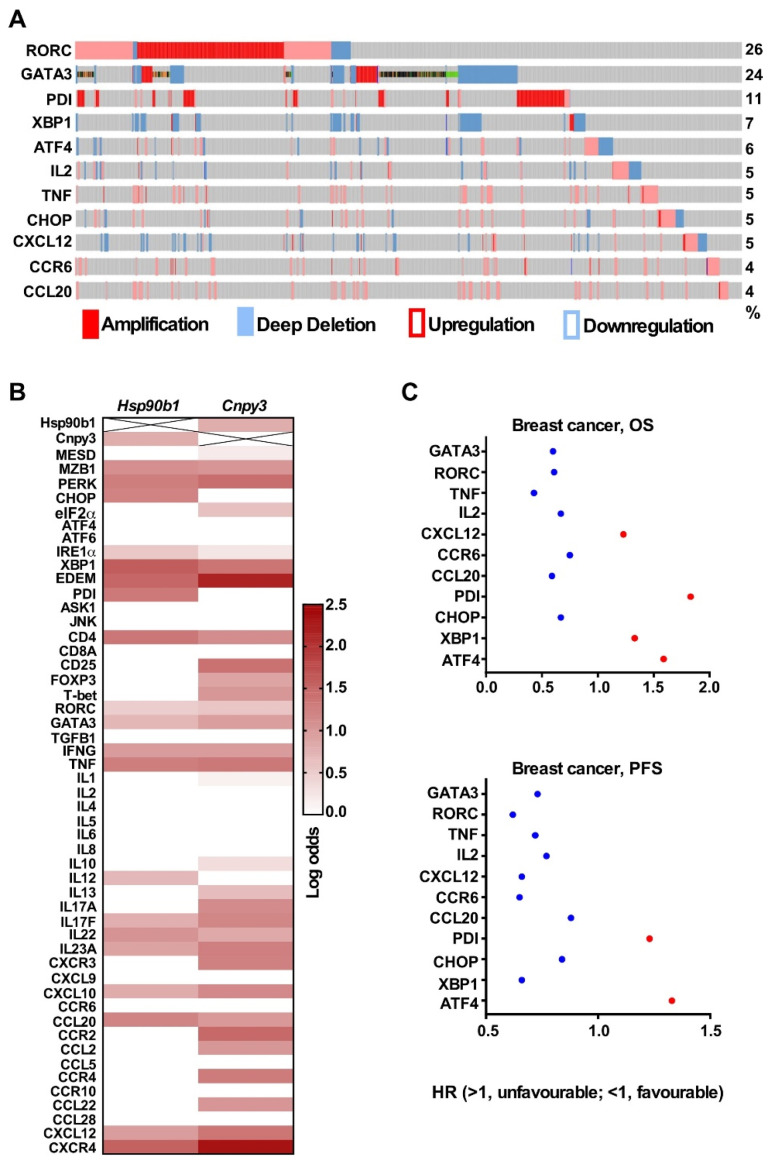
**CNPY3 and GRP94 alterations regulate UPR pathway and immune-related genes in the tumor microenvironment.** (**A**) cBioportal data show the gene alternation of enrichment genes based on the *HSP90B1* alternation, including the UPR pathway *ATF4*, *CHOP*, *XBP1* and *PDI* genes; Th17 and Th2 cell markers *RORC* and *GATA3*; and cytokines *CCL20*, *CCR6*, *CXCL12*, *IL2* and *TNF*. Red solid bars indicate gene amplifications, blue solid bars are deletions, and red blank bars indicate upregulation, and blue blank bars indicate downregulation. (**B**) Pair wise association plot for *HSP90B1*, *CNPY3*, the UPR pathway and immune-related genes based on the Curtis breast cancer dataset. This map provides summary statistics on mutual exclusivity and co-occurrence of genomic alterations in each pair of genes. The color scale represents the magnitude of the association (log odds). (**C**) Kaplan–Meier plotter data (http://kmplot.com, accessed on 15 October 2025) show the prognostic correlation of enrichment genes in breast cancer. Red dot (hazard ratio, HR > 1, *p* < 0.05), high expression is an unfavorable factor; blue dot (hazard ratio, HR < 1, *p* < 0.05), high expression is a favorable factor. OS, overall survival; PFS, progression-free survival.

## Data Availability

The original contributions presented in this study are included in the article/Appendix A. Further inquiries can be directed to the corresponding author.

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
