# Peer review of "CNPY3 Promotes Human Breast Cancer Progression and Metastasis via Modulation of the Tumor Microenvironment"

_cimb, 2025, doi:10.3390/cimb47110883_

Round 1

Reviewer 1 Report

Comments and Suggestions for Authors

In this work by Duan et al entitled “CNPY3 Promotes Human Breast Cancer Progression and Metas-tasis via Modulation of the Tumor Microenvironment”, the authors aim to shed new lights into the expression of CNPY3 in cancer development, looking mainly at breast cancer in this work. They interrogate different databases to establish some possible correlation between expression of this protein, along with GRP94 in carcinogenesis in different tissues, including that of the breast. Some work is then carried out using breast cancer cell lines to link expression of CNPY3 and colony formation/metastasis in vitro and in vivo. Further work is then presented to try to correlate CNPY3 changes to UPR pathways and tumour environment

Overall, this is an interesting piece of work which offers some analysis about the possible roles of CNPY3 in carcinogenesis. The majority of the work presented is however the result of online data analysis and as such, the work presents only coincidental correlation between changes in expression of their target protein and the different outcomes. Efforts should be made to clearly emphasis this important point. Sentences such as “CNPY3, a co-chaperone of GRP94, plays important roles in regulating the tumour microenvironment and anti-tumour immune response” and to a lesser extend “These results suggested that CNPY3 may be critical in human cancer development and progression” (as examples) are very misleading as the authors offer no proof of this

Points to consider:

Whilst the authors report some evidence indicating changes in expression of CNPY3 and GRP94 during carcinogenesis and specifically breast cancer using online databases, they do not provide any data correlating the two. It would be interesting to determine if there is coincidental expression of both markers simultaneously.

It is not clear if the creation of the CNPY3 knock out is a single clonal expansion or a pool collection. If the latter, do the authors see differential expression of the target proteins in the different cell population. If the former, are the result presented specific to this single clone or have others been tested to show the reproducibility of their work.

Figure 4 shows some interesting data but fails to be comprehensive. Two cells lines are presented, MCF-7 and MDA_MB-231) but only the latter is taken forward/used for the in vivo work. Why presenting only half the story without the MCF-7 cells?

The Kaplan Meier data analysis (Figure 4D) for animal survival with CNPY3 expression is rather surprising as the data shows a reduction in survival probability from 60% to 0 in a matter of hours/day. This is somewhat difficult to comprehend and would need to be clearly explained.

The work related to CNPY3, GRP94 and HSP90B1 is not clear and it is very difficult to understand the continuity of the work. Why introduce HSP90B1 now when none of the previous experiment considered this factor? Should not CNPY3 be the focus of the work?

Author Response

Dear reviewer 1, 

We thank you for you constructive critiques of our manuscript. We have revised the manuscript based on the suggestions and comments and addressed the questions (all changes are in red). A point-by-point response to reviewers' comments is described below.

"Sentences such as “CNPY3, a co-chaperone of GRP94, plays important roles in regulating the tumour microenvironment and anti-tumour immune response” and to a lesser extend “These results suggested that CNPY3 may be critical in human cancer development and progression” (as examples) are very misleading as the authors offer no proof of this."

Response: We apologize for the oversight, and thank you for the suggestions. We have revised the manuscript.

"Whilst the authors report some evidence indicating changes in expression of CNPY3 and GRP94 during carcinogenesis and specifically breast cancer using online databases, they do not provide any data correlating the two. It would be interesting to determine if there is coincidental expression of both markers simultaneously."

Response: Thank you for this important point. We agree with the reviewer. Determining whether CNPY3 and GRP94 are co-expressed would be a valuable line of inquiry for understanding their potential functional relationship in breast carcinogenesis. Our previous study has demonstrated that GRP94 forms a complex with CNPY3, and they are co-localized in the endoplasmic reticulum (Liu et al., Nature Communications, 2010). Our initial study was focused on establishing their individual expression levels and prognostic significance. We have shown that CNPY3 and GRP94 are both highly expressed in cancers (Figs. 1, 3, S1, S2), which indirectly revealed that CNPY3 and GRP94 are expressed in cancers simultaneously. We acknowledge this limitation and will examine both CNPY3 and GRP94 expression in cancer cells simultaneously in the future.

"It is not clear if the creation of the CNPY3 knock out is a single clonal expansion or a pool collection. If the latter, do the authors see differential expression of the target proteins in the different cell population. If the former, are the result presented specific to this single clone or have others been tested to show the reproducibility of their work."

 Response: Thank you for this comment. We apologize for the confusion. We used the CRISPR/Cas9 system to establish the CNPY3 knockout cell lines. We did not clone the CNPY3 KO cells, and both the MCF-7 and MDA-MB-231 breast cancer cell lines are pool KO cells.

"Figure 4 shows some interesting data but fails to be comprehensive. Two cells lines are presented, MCF-7 and MDA_MB-231) but only the latter is taken forward/used for the in vivo work. Why presenting only half the story without the MCF-7 cells?"

 Response: Thank you for this comment. We showed that the deletion of CNPY3 suppressed both MCF-7 and MDA-MB-231 human breast cancer cells. However, the MDA-MB-231 cell line is a poorly differentiated triple-negative breast cancer and has a stronger invasive ability than MCF-7, and it is a well-known metastatic breast cancer model. In order to investigate the role of CNPY3 in tumor progression and metastasis, we used the MDA-MB-231 tumor model for in vivo experiments.

"The Kaplan Meier data analysis (Figure 4D) for animal survival with CNPY3 expression is rather surprising as the data shows a reduction in survival probability from 60% to 0 in a matter of hours/day. This is somewhat difficult to comprehend and would need to be clearly explained."

 Response: Thank you for this comment. The survival data in Figure 4D represents an endpoint survival when the mice reached the humane endpoint based on our animal protocol approved by OSU IACUC. If the tumor size reaches 1.6 cm or the tumor mass reaches 3.2 cubic cm, or a combination of signs (lethargy, decreased mobility, and a hunched posture) occurs, the mice will be euthanized based on the early removal criteria as an end point. We have included this information in the Method section.

"The work related to CNPY3, GRP94 and HSP90B1 is not clear and it is very difficult to understand the continuity of the work. Why introduce HSP90B1 now when none of the previous experiment considered this factor? Should not CNPY3 be the focus of the work?"

 Response: Thank you for this comment. First, we would like to clarify that GRP94 is the protein product of the HSP90B1 gene. The central focus of this work is CNPY3. Our previous study has demonstrated that CNPY3 is a co-chaperone of GRP94 and critical for the maturation and signaling of TLRs. GRP94 is an essential chaperone for integrins, LRP6, GARP, etc., which are involved in tumor initiation and progression. However, it is unclear whether CNPY3 plays a similar role as GRP94 in cancers. Thus, we logically investigated whether GRP94 is also dysregulated in parallel. In the Introduction, we have clearly stated that GRP94 is encoded by the HSP90B1 gene and have established the well-documented biological relationship between CNPY3 and GRP94, providing a rationale for studying them together.

Reviewer 2 Report

Comments and Suggestions for Authors

This study provides a comprehensive investigation into the role of CNPY3 as a co-chaperone of GRP94 in promoting breast cancer progression and metastasis through modulation of the tumor microenvironment. The authors utilized multi-omics data, clinical samples, and functional experiments to demonstrate that CNPY3 is overexpressed in breast cancer and associated with poor prognosis. CRISPR/Cas9-mediated knockout of CNPY3 suppressed tumor growth and metastasis in vitro and in vivo. The study is well-structured and provides valuable insights into CNPY3 as a potential therapeutic target. However, several points require clarification or further validation. Specific comments are as follows.
1. While the use of NSG mice and xenograft models is appropriate for assessing tumor growth and metastasis, the study lacks a rationale for selecting MCF-7 (ER+/PR+) and MDA-MB-231 (TNBC) cells in the context of CNPY3’s role across breast cancer subtypes. It would be beneficial to include additional cell lines representing HER2+ or other subtypes to strengthen the generalizability of the findings.
2. The study suggests that CNPY3 and GRP94 regulate UPR pathway genes and immune-related genes in the TME, supported largely by bioinformatic analyses (e.g., Oncomine, cBioPortal). However, functional validation of these pathways—such as rescue experiments or pharmacological inhibition of UPR components—is lacking. Including such experiments would strengthen the mechanistic claims.
3. The weak correlation between CNPY3 and GRP94 mRNA levels (Figure 5C) contrasts with their proposed functional synergy. Further investigation into whether CNPY3’s oncogenic effects are fully dependent on GRP94—e.g., through double knockdown or rescue with GRP94—would clarify their relationship.
4. Although Figure 4E shows a reduction in metastasis upon CNPY3 knockout, the method of quantification (e.g., imaging, histology) is not detailed. Providing representative images or more detailed metastatic burden analysis (e.g., number of metastatic nodules, organ-specific metastasis) would enhance the credibility of these findings.
5. The study highlights CNPY3 as a potential biomarker and therapeutic target. However, no discussion is provided regarding how CNPY3 expression might be targeted therapeutically (e.g., small molecule inhibitors, antibodies) or how it could be integrated into existing breast cancer treatment paradigms.
6. While Figures S1 and S2 and Table S1 provide supportive data, their relevance to the main conclusions could be better integrated into the main text. For example, the elevated expression of CNPY3 in other cancers (Figure S1) could be discussed in the context of pan-cancer potential.

Author Response

Dear reviewer 2, 

We thank you for you constructive critiques of our manuscript. We have revised the manuscript based on the suggestions and comments and addressed the questions (all changes are in red). A point-by-point response to reviewers' comments is described below.

"While the use of NSG mice and xenograft models is appropriate for assessing tumor growth and metastasis, the study lacks a rationale for selecting MCF-7 (ER+/PR+) and MDA-MB-231 (TNBC) cells in the context of CNPY3’s role across breast cancer subtypes. It would be beneficial to include additional cell lines representing HER2+ or other subtypes to strengthen the generalizability of the findings."

Response: Thank you for the comments and suggestions. We showed that the deletion of CNPY3 suppressed both ER+ MCF-7 and triple-negative MDA-MB-231 human breast cancer cells (Figure 4). However, the MDA-MB-231 cell line is a poorly differentiated triple-negative breast cancer (TNBC) and has a stronger invasive ability than MCF-7, and it is a well-known metastatic breast cancer model. In order to investigate the role of CNPY3 in tumor progression and metastasis, we used the MDA-MB-231 tumor model for in vivo experiments. The expression level of CNPY3 is strongly associated with TNBC and basal subtypes (Figs. 1D and 1F), suggesting CNPY3 subtype-specific roles. Further studies comparing CNPY3 dependency across molecular subtypes are warranted in the future.

"The study suggests that CNPY3 and GRP94 regulate UPR pathway genes and immune-related genes in the TME, supported largely by bioinformatic analyses (e.g., Oncomine, cBioPortal). However, functional validation of these pathways—such as rescue experiments or pharmacological inhibition of UPR components—is lacking. Including such experiments would strengthen the mechanistic claims."

Response: We strongly agree with the reviewer and acknowledge this limitation. While our study focused on establishing CNPY3’s pro-tumorigenic role and its correlation with UPR/immune genes, direct mechanistic validation in TME requires further investigation. We have collaborated with Dr. Andrew Hu, who has developed small inhibitors for blocking XBP1s in vivo. We plan to perform further study to investigate the role of CNPY3-UPR signaling in vivo using XBP1s inhibitors and genetic approach.

"The weak correlation between CNPY3 and GRP94 mRNA levels (Figure 5C) contrasts with their proposed functional synergy. Further investigation into whether CNPY3’s oncogenic effects are fully dependent on GRP94—e.g., through double knockdown or rescue with GRP94—would clarify their relationship."

Response: Thank you for the suggestions. We agree with the reviewer. The mRNA levels might not be the same as protein levels. We will further investigate whether CNPY3 oncogenic effects are dependent on GRP94 by knocking down GRP94 in CNPY3 KO cells or treating them with a GRP94-specific inhibitor that our collaborator Dr. Chiosis from MSKCC has developed.

"Although Figure 4E shows a reduction in metastasis upon CNPY3 knockout, the method of quantification (e.g., imaging, histology) is not detailed. Providing representative images or more detailed metastatic burden analysis (e.g., number of metastatic nodules, organ-specific metastasis) would enhance the credibility of these findings."

Response: Thank you for this comment and suggestion. Our human breast cancer cells do not have reporters. Since the tumor cells were implanted into the fourth mammary gland of NSG mice, most tumors metastasized to the liver and lymph nodes. At the end point, we examined metastatic tumors in the liver and lymph nodes under a dissecting microscope.   

"The study highlights CNPY3 as a potential biomarker and therapeutic target. However, no discussion is provided regarding how CNPY3 expression might be targeted therapeutically (e.g., small molecule inhibitors, antibodies) or how it could be integrated into existing breast cancer treatment paradigms."

Response: We agree with the reviewer. Our study, for the first time, showed that CNPY3 is highly expressed in breast cancers, and the deletion of CNPY3 decreased tumor progression, suggesting CNPY3 might be a potential target for cancer treatment and enhance checkpoint blockade immunotherapy in TNBC that is resistant to immunotherapy. Currently, there are no small molecule inhibitors and antibodies for targeting CNPY3. We will collaborate with our drug screening core and antibody core to develop CNPY3-specific inhibitors and antibodies for preclinical study. We have discussed that CNPY3 might be a potential target for therapeutics in the Discussion section.

"While Figures S1 and S2 and Table S1 provide supportive data, their relevance to the main conclusions could be better integrated into the main text. For example, the elevated expression of CNPY3 in other cancers (Figure S1) could be discussed in the context of pan-cancer potential."

Response: Thank you for the comments and suggestions. We have revised the manuscript and discussed supplemental data in the Discussion section.

Round 2

Reviewer 1 Report

Comments and Suggestions for Authors

Following the first review, the authors have made some text amendments in the manuscript to provide more background and explanations. Their answers however fail to address some of the initial points that were raised and are still unresolved as listed below:

To the question related to Figure 4 and that two cells lines are presented, MCF-7 and MDA_MB-231) but only the latter is taken forward/used for the in vivo work. Why presenting only half the story without the MCF-7 cells? The authors respond by informing us that “MDA-MB-231 cell line is a poorly differentiated triple-negative breast cancer and has a stronger invasive ability than MCF-7, and it is a well-known metastatic breast cancer model”. This argument then suggests that experiments with MCF7 may not bring the same outcome. This would be valuable to confirm as currently the work does not clearly show why one of the cell lines which is presented initially is not taken forward for the remaining of the experiments. Why presenting the MCF7 data at the start if the cell line is not suitable for your analysis?

Authors have now informed us that “MCF-7 and MDA-MB-231 breast cancer cell lines are pool KO cells” following CNPY3 knock down. The question raised initially still stands in that, do the authors see differential expression of the target proteins in the different cell population?

Reviewer 2 Report

Comments and Suggestions for Authors

The authors have addressed the problem very well, and the manuscript can be accepted in the present form.

Author Response

"The authors have addressed the problem very well, and the manuscript can be accepted in the present form."

Response: Thank you for the valuable comments throughout the review process. We are very pleased that the reviewer has recommended acceptance.

Round 3

Reviewer 1 Report

Comments and Suggestions for Authors

Suitable changes have been made